# Antibody Response to the SARS-CoV-2 Spike and Nucleocapsid Proteins in Patients with Different COVID-19 Clinical Profiles

**DOI:** 10.3390/v15040898

**Published:** 2023-03-31

**Authors:** Sinei Ramos Soares, Maria Karoliny da Silva Torres, Sandra Souza Lima, Kevin Matheus Lima de Sarges, Erika Ferreira dos Santos, Mioni Thieli Figueiredo Magalhães de Brito, Andréa Luciana Soares da Silva, Mauro de Meira Leite, Flávia Póvoa da Costa, Marcos Henrique Damasceno Cantanhede, Rosilene da Silva, Adriana de Oliveira Lameira Veríssimo, Izaura Maria Vieira Cayres Vallinoto, Rosimar Neris Martins Feitosa, Juarez Antônio Simões Quaresma, Tânia do Socorro Souza Chaves, Giselle Maria Rachid Viana, Luiz Fábio Magno Falcão, Eduardo José Melo dos Santos, Antonio Carlos Rosário Vallinoto, Andréa Nazaré Monteiro Rangel da Silva

**Affiliations:** 1Laboratório de Virologia, Instituto de Ciências Biológicas, Universidade Federal do Pará, Belém 66075-110, Brazilvallinoto@ufpa.br (A.C.R.V.); 2Programa de Pós-Graduação em Biologia de Agentes Infecciosos e Parasitários, Universidade Federal do Pará, Belém 66075-110, Brazil; 3Laboratório de Genética de Doenças Complexas, Instituto de Ciências Biológicas, Universidade Federal do Pará, Belém 66075-110, Brazil; 4Hospital Adventista de Belém, Belém 66093-904, Brazil; 5Centro de Ciências Biológicas e da Saúde, Universidade do Estado do Pará, Belém 66050-540, Brazil; 6Laboratório de Pesquisas Básicas em Malária em Malária, Seção de Parasitologia, Instituto Evandro Chagas, Secretaria de Ciência, Tecnologia e Insumos Estratégicos, Ministério da Saúde do Brasil, Ananindeua 70068-900, Brazil

**Keywords:** SARS-CoV-2, immunity, IgG antibodies, COVID-19

## Abstract

The first case of coronavirus disease 2019 (COVID-19), caused by severe acute respiratory syndrome coronavirus 2 (SARS-CoV-2), in Brazil was diagnosed on February 26, 2020. Due to the important epidemiological impact of COVID-19, the present study aimed to analyze the specificity of IgG antibody responses to the S1, S2 and N proteins of SARS-CoV-2 in different COVID-19 clinical profiles. This study enrolled 136 individuals who were diagnosed with or without COVID-19 based on clinical findings and laboratory results and classified as asymptomatic or as having mild, moderate or severe disease. Data collection was performed through a semistructured questionnaire to obtain demographic information and main clinical manifestations. IgG antibody responses to the S1 and S2 subunits of the spike (S) protein and the nucleocapsid (N) protein were evaluated using an enzyme-linked immunosorbent assay (ELISA) according to the manufacturer’s instructions. The results showed that among the participants, 87.5% (119/136) exhibited IgG responses to the S1 subunit and 88.25% (120/136) to N. Conversely, only 14.44% of the subjects (21/136) displayed S2 subunit responses. When analyzing the IgG antibody response while considering the different proteins of the virus, patients with severe disease had significantly higher antibody responses to N and S1 than asymptomatic individuals (*p* ≤ 0.0001), whereas most of the participants had low antibody titers against the S2 subunit. In addition, individuals with long COVID-19 showed a greater IgG response profile than those with symptomatology of a short duration. Based on the results of this study, it is concluded that levels of IgG antibodies may be related to the clinical evolution of COVID-19, with high levels of IgG antibodies against S1 and N in severe cases and in individuals with long COVID-19.

## 1. Introduction

The emergence of a novel human coronavirus (severe acute respiratory syndrome coronavirus 2, SARS-CoV-2) at the end of 2019, which quickly spread to all Chinese provinces and to more than 100 countries on all continents [1,2], led the World Health Organization (WHO) to consider and define the situation as a new pandemic. SARS-CoV-2 is the seventh coronavirus known to infect humans. SARS-CoV, Middle East respiratory syndrome coronavirus (MERS-CoV), and SARS-CoV-2 can cause severe illness, whereas HKU1, NL63, OC43, and 229E are associated with mild disease [3].

The SARS-CoV-2 genome is approximately 30 kb, encoding four structural proteins (S, E, M and N) and sixteen nonstructural proteins (nsp1-nsp16) [4]. The N protein (nucleocapsid) forms the capsid that encloses the genome; the other structural proteins S (spike), E (envelope) and M (membrane) are associated with the viral envelope [5].

The S glycoprotein is responsible for binding to the cell receptor ACE2 (angiotensin-converting enzyme 2). Spike forms a homotrimer that projects from the viral surface and comprises two subunits: S1 and S2. The S1 subunit is responsible for binding to the cellular receptor from its distal region, which contains the receptor-binding domain (RBD); this contributes to stabilization of the prefusion state of the S2 subunit, which is involved in fusion of the viral envelope with the cell membrane [6].

Because the S protein is exposed on the surface of the virus, it has been used as the main target antigen in vaccine development, especially due to the ability of the RBD to elicit neutralizing antibody and T-cell responses [7]. However, the vaccine response may be impacted by the similarity of amino acids between the spike proteins of SARS-CoV-2 and SARS-CoV and the possibility of nonsynonymous mutations in the S gene. The N gene, which encodes nucleoprotein, undergoes fewer mutations over time [8].

The N protein of many coronaviruses is highly immunogenic and abundantly expressed during infection. High levels of IgG antibodies against N have already been detected in sera from patients infected with SARS-CoV. With regard to SARS-CoV-2 infection, convalescent individuals seem to have specific T cells for the N protein [9,10]. Because they elicit a directed humoral and cellular response, the structural proteins S and N have been used as target antigens in serological assays for SARS-CoV-2 [11].

SARS-CoV-2 infects the upper and lower respiratory tract and can cause a respiratory syndrome that varies among patients in duration and severity. The clinical symptoms of coronavirus disease 2019 (COVID-19) are similar to those of influenza, with many individuals presenting an asymptomatic state and others developing a more severe and critical clinical condition requiring oxygen therapy and ventilation support [12,13]. In addition, some individuals with acute SARS-CoV-2 infection develop a wide range of persistent symptoms that do not resolve over many months, which is called long COVID-19 syndrome [14]. Viral proteins and their interactions with host factors play a key role in imbalanced immune responses, with synthesis of high levels of proinflammatory cytokines that have a direct impact on disease severity [15,16].

One of the best ways to prevent COVID-19 is by establishing high levels of herd immunity through infection or vaccination. However, herd immunity by infection appears to difficult to achieve for COVID-19 because reinfection may occur; vaccine development should also be improved for better efficacy and to manage highly contagious variants. The current vaccines were developed using S as the immunogen, but it has been observed that an immune response can be mounted to different antigens of the virus [17,18]. Furthermore, although it remains unclear which factors cause some individuals to have more severe disease than others, it may be related to a different profile of immune response to different SARS-CoV-2 proteins.

Thus, the present study aimed to understand how the humoral response to different proteins of SARS-CoV-2 occurs. The findings will contribute to decision-making for the development of future vaccines better directed toward viral targets (epitopes), to our understanding of the clinical status of individuals and to identifying a prognostic marker.

## 2. Materials and Methods

### 2.1. Study Design and Sampling

The subjects for this research were selected from among individuals who had a clinical and laboratory diagnosis (RT-PCR) of COVID-19, with disease classified according to severity as mild, moderate or severe. The sample collection was from June 2020 to June 2021. The patients underwent treatment at home or at Adventist Hospital of Belém and in the Post-COVID Clinic of the University of the State of Pará (UEPA).

The individuals also underwent an interview in which they answered questions through a semistructured questionnaire that compiled information on demographic aspects and social behavior during the pandemic. A total of 136 individuals who had confirmed infection and clinical signs of the disease were selected. The cases were classified as mild (*n* = 33), moderate (*n* = 33) or severe (*n* = 33) COVID-19 following the criteria of the WHO [19] with regard to the occurrence of clinical symptoms of fever, cough, fatigue, anorexia, shortness of breath, myalgia, pneumonia, acute respiratory distress syndrome and oxygenation impairment. Individuals confirmed to be positive for the virus but who were asymptomatic for the disease (*n* = 37) were also recruited. Among the participants, those who maintained symptoms (sequelae) associated with COVID-19 for 30 days after resolution of the acute condition were diagnosed with long COVID-19 syndrome (*n* = 59), forming a group for comparisons with acute COVID-19 patients (*n* = 33). The main symptoms reported in the long COVID-19 group were headache, depression, anxiety, insomnia, tingling in extremities, mild cognitive disorder, tiredness, dyspnea, weakness, pain, anosmia, and ageusia.

The sample consisted of individuals of both sexes aged older than 18 years who signed informed consent, from whom a biological sample (blood) was collected and who agreed to respond to the epidemiological questionnaire. Blood samples were collected on average at 120 days after the manifestation of symptoms, and the individuals had not been vaccinated against COVID-19. For those diagnosed with long COVID-19, a blood sample was collected at approximately 90 days after resolution of acute COVID-19. Individuals who did not agree to answer the epidemiological questionnaire and who reported being vaccinated against COVID-19 were excluded from the research. The present study followed the Regulatory Guidelines and Norms for Research Involving Human Subjects (CAE: 33470020.0.1001.0018).

### 2.2. Analysis of IgG Responses to SARS-CoV-2 S1, S2 and N Proteins

Peripheral blood (10 mL) was collected into EDTA tubes using a venipuncture collection system. Plasma was separated and used for analysis; 136 samples were tested to determine the specificity of IgG antibodies against the immunodominant S1 and S2 subunits and N protein of SARS-CoV-2. A COVID-19 IgG Confirmation kit (Dia. Pro-Diagnostic Bioprobes, Milan, Italy) was used according to the manufacturer’s procedures. Briefly, the test strips of the kit contain BSA (lanes 1, 5 and 9-negative test control), S1 (lanes 2, 6 and 10-recombinant spike 1 glycoprotein), S2 protein (lanes 3, 7 and 11-recombinant spike 2 glycoprotein) and C (lanes 4, 8 and 12-nucleoprotein, core). Diluted samples were dispensed horizontally along the module strips in wells 1–4 and incubated for 45 min at 37 °C. After washing the sample components, bound antibodies were detected by the addition of anti-IgG antibodies labeled with peroxidase (HRP). The enzyme captured in the solid phase, acting on the substrate/chromogen, generates an optical signal that is proportional to the amount of IgG present in the sample.

The presence of IgG in each sample was semiquantitatively determined by the mean of the cutoff value capable of discriminating between positive and negative samples. The parameters provided by the kit were used to assess the IgG response (named the neutralization profile by the manufacturers), which considers the test results for IgG Spike subunit 1 (S1) and interprets from the OD of S/Co values, according to the following (Co = Cut-Off values): Low Potential Neutralizing Efficacy (Range 2 < S/Co <4), Medium Potential Neutralizing Efficacy (Range 4 ≤ S/Co < 8, IgG Titer 1:80–1:160), High Potential Neutralizing Efficacy (Range 8 ≤ S/Co < 12, IgG Titer 1:160–1:240), Very High Potential Neutralizing Efficacy (Range ≥ 12, IgG Titer > 1:240).

### 2.3. Statistical Analysis

To assess the association between the clinical profile of individuals and response to SARS-CoV-2 proteins, the Kruskal-Wallis test was performed on optical density (OD) values. The G test and Kruskal-Wallis test were used to analyze the association between the clinical profile of individuals and response to SARS-CoV-2 proteins according to the effectiveness of the IgG antibody response (low, medium, high). The statistical analyses were performed using the RStudio program version 4.1.1 and GraphPad version 8.0. A significance level of 5% was adopted.

## 3. Results

### 3.1. Clinical-Demographic Characteristics

The main clinical-demographic aspects of the groups of patients investigated in the present study are shown in Table 1. There were no significant differences between groups regarding sex or age. However, symptoms (cough, chest pain, abdominal pain, shortness of breath), hospitalization, ICU admission and ventilatory support were significantly more frequent in the moderate and severe groups.

### 3.2. Anti-N, -S1 and -S2 Protein IgG Antibody Responses

Most of the participants exhibited a profile of positive responses of IgG antibodies to the SARS-CoV-2 S1 subunit and N. The results showed that 87.5% of the participants (119/136) responded to the S1 subunit and that 88.25% (120/136) responded to N. For the S1 subunit, 96.97% of the individuals with severe clinical disease (32/33), 87.88% of those with moderate disease (29/33), 93.94% of those with mild disease (31/33), and 72.97% of the asymptomatic individuals (27/37) displayed a response (Figure 1A). Only 14.44% of the individuals tested positive for specific IgG (21/136) against the S2 subunit (Figure 1B). The results for the N protein were similar to those for the S1 subunit, with IgG positivity detected for 96.97% (32/33) of individuals with severe disease, 87.88% (29/33) of those with moderate disease, 96.97% (32/33) of those with mild disease, and 72.97% (27/37) of asymptomatic individuals (Figure 1C).

### 3.3. The IgG Antibody Response According to the Clinical Profile

When analyzing the IgG antibody response with regard to the different proteins of the virus, the group of severe patients showed higher antibody responses to S1 and N than the group of asymptomatic individuals (Table 2; Figure 2A,C). For the S1 subunit, most individuals with severe disease (72.73%) had a high IgG response; however, this response occurred in only 10.81% of asymptomatic individuals. Moreover, only a minority of individuals with severe COVID-19 (6.06%) had low titers of IgG antibodies. Overall, asymptomatic individuals showed medium levels of IgG antibodies (35.14%).

Although severe individuals exhibited a high IgG response to the S2 subunit, it is noteworthy that only 2 patients in this group exhibited IgG responses to S2. No significant difference was observed when comparing the different groups (Table 2; Figure 2B; *p* = 0.1478).

In terms of the IgG antibody response, the results for the N protein (Table 2; Figure 2C) were similar to those for the S1 subunit, with most individuals with severe COVID-19 (69.7%) showing the highest responses when compared with the asymptomatic group (8,1%). Asymptomatic individuals mostly exhibited lower levels of an IgG antibody response (32.4%).

Regarding the S1 subunit (Figure 2A), a statistically significant difference was found between asymptomatic individuals and those with severe COVID-19 (*p* < 0.0001), as well as between the mild and severe COVID-19 groups (*p* = 0.0001) and between the severe and moderate COVID-19 groups (*p* = 0.0169). Conversely, there was no significant difference between the groups analyzed with respect to the S2 subunit (Figure 2B). When examining the N protein (Figure 2C), there was a statistically significant difference (*p* < 0.0001) between individuals in the asymptomatic and severe disease groups and between individuals in the mild and severe disease groups (*p* = 0.0226), but there was no significant difference between the severe and moderate disease groups (*p* = 0.2922).

When analyzing the IgG response to the different proteins tested and their association with acute COVID-19 and long COVID-19, it was observed that those who developed the latter displayed a greater response to S1 and N, with a significant difference between the groups (*p* = 0.003 for S1 and *p* = 0.0085 for N), as illustrated in Figure 3A,B. There was no significant difference between the groups for S2 (*p* = 0.9189).

## 4. Discussion

In our study, we verified the presence of anti-SARS-CoV-2 IgG antibody immune responses against the S S1 subunit and N protein. Similar results to ours were found in another study that investigated the antibody repertoires of hundreds of patients with COVID-19 and a control group (samples collected in the pre-COVID-19 period) and identified hundreds of antibody targets, which recognized mainly S and N in the case of infected individuals [20].

The N protein is the most abundant viral structural protein, and similar to the S1 subunit, it is the most immunogenic protein of the virus particle, being able to bind to viral RNA to form the core of the ribonucleoprotein complex [21,22]. When comparing the IgG antibody response to SARS-CoV-2 in infected patients versus controls, most COVID-19 patients exhibit a strong response, with IgG antibodies against N and, to a lesser extent, a fragment containing amino acids 300–685 of S [23].

Nonetheless, our findings indicated little response to the S2 subunit; although it contains regions conserved among coronaviruses, S2 may contain epitopes targeted by broadly neutralizing antibodies that are less specific than the S1 RBD. In general, immunodominant sites need to be better elucidated to improve the sensitivity of diagnostic tests [24].

In our study, the association between the different clinical profiles of individuals and the levels of IgG of anti-SARS-CoV-2 antibodies was significantly greater in severe cases, with a high anti-SARS-CoV-2 IgG antibody response to S1 and N. These results corroborate a study comparing the performance of S and N protein antigens for detecting IgG, IgM and IgA antibodies specific for SARS-CoV-2, with high levels of neutralizing antibodies in severe and mild cases [25]. It has also been reported that antibodies against SARS-CoV-2 are produced at high levels in patients with severe symptoms after infection [26]. One study reported that asymptomatic infection elicits weaker antibody responses than symptomatic infection and induces primarily IgG-based antibody responses [27]. Another study analyzed 343 patients infected with SARS-CoV-2 and found that IgG antibody responses to the RBD region correlated with anti-S neutralizing antibody titers, with little or no decrease in titers over 75 days from the onset of symptoms [28].

In our previous report [29], IgG against SARS-CoV-2 was lost in 30% of patients at 90 days after diagnosis, the frequency of all reported symptoms was higher in individuals who maintained IgG persistence after 90 days from symptom onset, and the length of hospital stay and supplemental oxygen use were higher in individuals with a persistent IgG response. To avoid the influence of sampling time on levels of antibody responses observed in the present study, we analyzed antibody levels between asymptomatic, mild, moderate and severe groups while accounting for intervals of sample collection (less than and greater than 90 days), and the results showed no significant differences (data not shown).

As with our results, IgG antibody titers correlate with disease severity [30]. Despite the indication of a successful neutralizing response in most individuals, higher titers may be associated with more severe clinical cases, suggesting that a robust antibody response alone is insufficient to prevent severe disease and that it may be harmful or even ineffective rather than protective. Evidence supports the existence of preexisting, nonneutralizing or poorly neutralizing antibodies developing as a result of vaccination or past infection, which may also contribute to more severe subsequent infection, a phenomenon called antibody-dependent enhancement (ADE) [31].

Indeed, one study involving the ADE effect on SARS-CoV-2 confirmed an increase in antibody dependence after COVID-19 [32]. Our findings reveal that SARS-CoV-2 infection leads to a robust IgG antibody response in individuals with a severe clinical profile, similar to what was observed in a study that reported high titers of antibodies against S in severe cases relative to mild or asymptomatic cases [33].

Long COVID-19 is a complex condition with diverse and prolonged symptoms [34]. Known as postacute COVID-19, the condition has also been diagnosed in asymptomatic individuals or those who had mild or severe disease [35]. The symptoms may be a consequence of tissue damage caused by the virus or the inflammation that occurs during acute COVID-19 [36]. Although long COVID-19 has been reported to be associated with a weak anti-SARS-CoV-2 antibody response and mild disease severity [37], the subjects with long COVID-19 in the present study were classified as having the clinical features of severe or moderate disease and higher antibody titers; in contrast, individuals with acute COVID-19 and lower antibody titers were characterized as having mild disease. This result is probably more related to severity during the acute COVID-19 period than to differences in response between long and acute COVID-19. Future analyses addressing the pairing of clinical status between groups may resolve this issue. From this perspective, it has already been reported that in individuals with severe disease, neutralizing antibodies are self-reactive and directed against the organs and tissues of the host [38]. In addition, SARS-CoV-2 may remain in some individuals in whom the virus is not adequately cleared, causing chronic inflammation and dysfunction in certain organs and tissues in severe cases [39].

Our study has some limitations, as (i) we did not screen the patients for the presence of coinfections, (ii) the data presented on the level of anti-S-1, anti-S-2, and anti-N IgG antibodies were deduced from antibody concentrations, as suggested by the test manufacturer, and (iii) we were unable to investigate the association of SARS-CoV-2 variants or antigenemia with anti-N, -S1 and -S2 protein IgG antibody responses. Thus, more studies are needed to elucidate the profile of neutralizing antibody responses in individuals with acute and long COVID-19 as well as markers that can be used for prognosis.

## 5. Conclusions

In this study, most of the individuals analyzed produced antibody responses against the S protein S1 subunit and the N protein. Individuals with a severe clinical profile showed the highest IgG response profile for the S1 subunit and N. Individuals with long COVID-19 had the highest antibody titers and were classified as having severe or moderate disease.

## Figures and Tables

**Figure 1 viruses-15-00898-f001:**
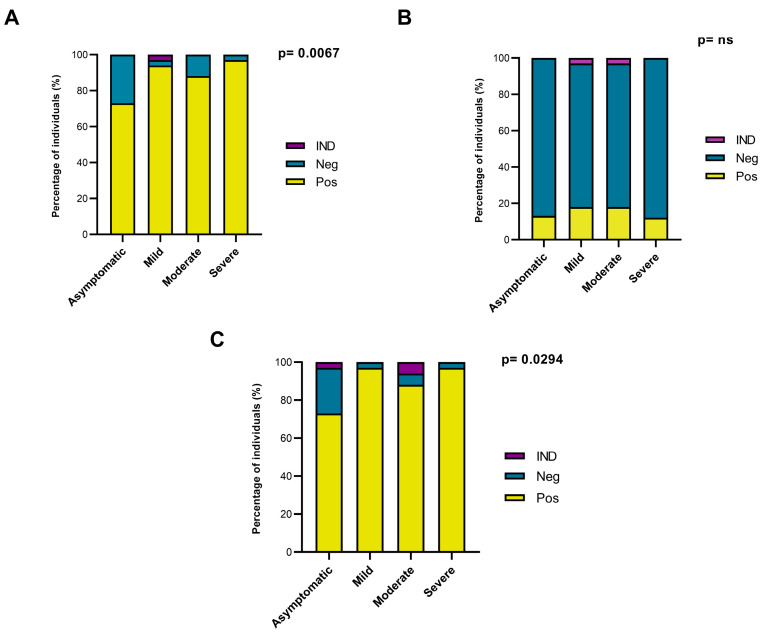
Analysis of the total IgG response to SARS-CoV-2 S and N proteins in individuals with different clinical profiles of COVID-19. (**A**) Analysis of the total IgG response to the S1 subunit of S. (**B**) Analysis of the total IgG response to the S2 subunit of S. (**C**) Analysis of the total IgG response to N. Pos: Positive ELISA result. Neg: Negative ELISA result. IND: Indeterminate ELISA result. ns: Non-significant.

**Figure 2 viruses-15-00898-f002:**
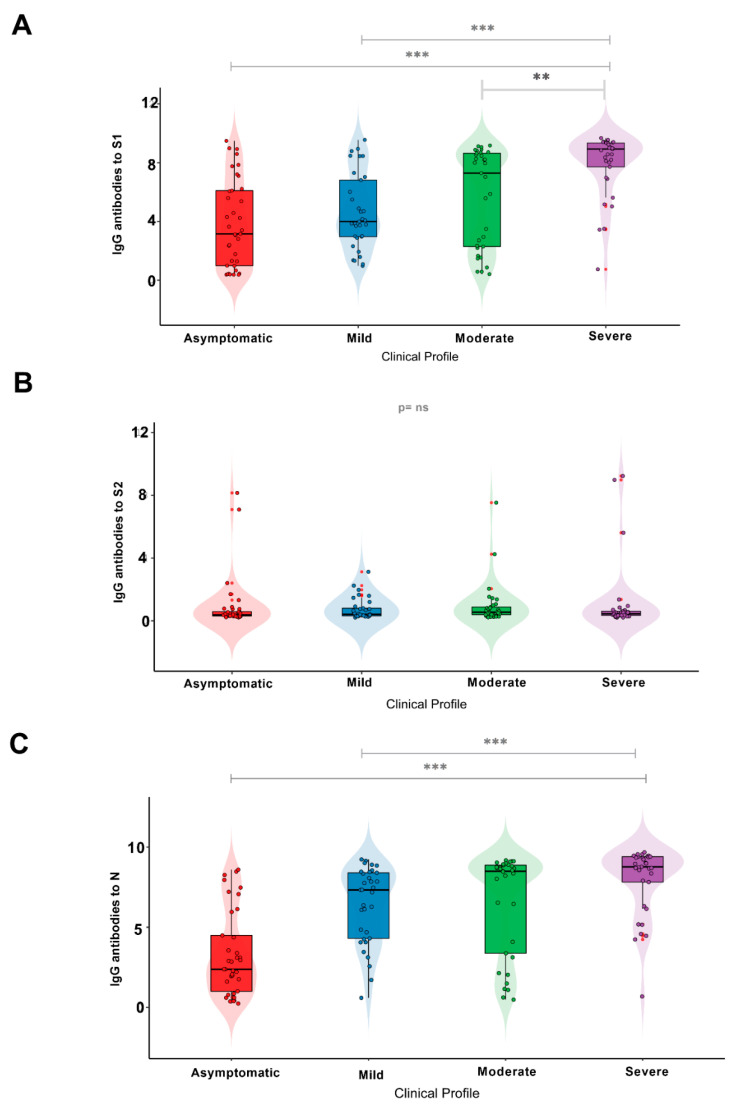
IgG antibody response to the (**A**) S1 and (**B**) S2 subunits and **(C)** N protein in individuals with different clinical profiles of COVID-19. Significance value *p* < 0.002 ** and *p* < 0.0001 ***. ns: Non-significant.

**Figure 3 viruses-15-00898-f003:**
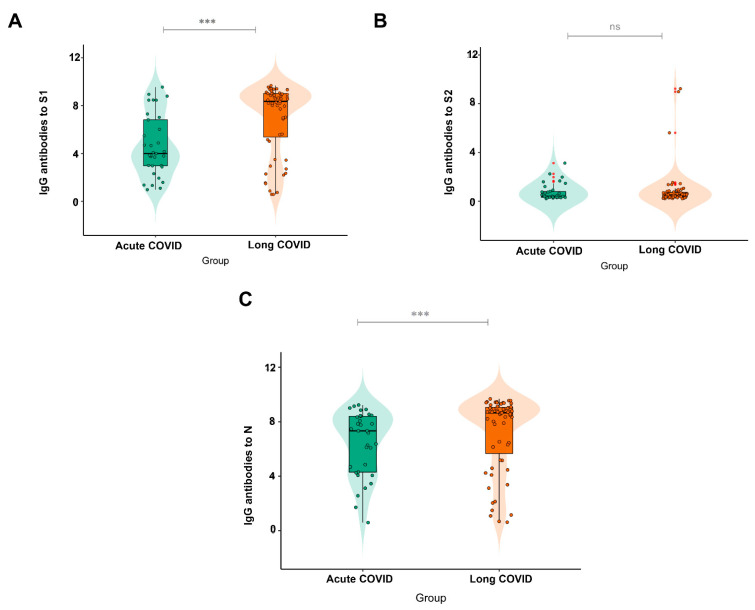
IgG antibody response to (**A**) (N = 56 Acute COVID; N = 30 Long COVID) the S1 and (**B**) (N = 13 Acute COVID; N = 8 Long COVID) S2 subunits and (**C**) (N = 59 Acute COVID; N = 31 Long COVID) the N protein in subjects with acute COVID-19 and long COVID-19. Significance value *p* < 0.0001 ***. ns: Non-significant.

**Table 1 viruses-15-00898-t001:** Clinical-demographic characteristics of patients infected with SARS-CoV-2.

Variables	Asymptomatic *n* = 37 (%)	Mild *n* = 33 (%)	Moderate *n* = 33 (%)	Severe *n* = 33 (%)	*p* Values
Sex					
Female	26 (70.3)	15 (45.5)	15 (45.5)	17 (51.5)	0.1139
Male	11 (29.7)	18 (54.5)	18 (54.5)	16 (48.5)	
Age					
Median	48.4	38.0	47.0	45.0	0.9165
Average	50.0	38.2	47.4	46.8	
SD	19.4	8.92	13.83	11.65	
Symptoms					
Fever	-	24 (72.72)	25 (75.75)	28 (84.84)	0.4678
Cough	-	14 (42.42)	24 (72.72)	26 (78.78)	0.0042
Runny nose	-	17 (51.51)	9 (27.27)	15 (45.45)	0.1148
Headache	-	18 (54.54)	22 (66.66)	24 (72.72)	0.2901
Sore throat	-	12 (36.36)	16 (48.48)	20 (60.60)	0.1435
Chest pain	-	11 (33.33)	14 (42.42)	24 (72.72)	0.0036
Abdominal pain	-	3 (9.09)	13 (39.39)	12 (36.36)	0.0108
Body ache	-	21 (63.63)	25 (75.75)	22 (66.66)	0.5431
Nausea	-	7 (21.21)	9 (27.27)	11 (33.33)	0.5427
Vomiting	-	2 (6.06)	7 (21.21)	5 (15.15)	0.1956
Diarrhea	-	11 (33.33)	19 (57.57)	16 (48.48)	0.1367
Shortness of breath	-	10 (30.30)	19 (57.57)	30 (90.90)	<0.0001
Weakness	-	17 (51.51)	13 (39.39)	10 (30.30)	0.1180
Tiredness	-	15 (45.45)	22 (66.66)	17 (51.51)	0.2042
Anosmia	-	22 (66.66)	19 (57.57)	19 (57.57)	0.6833
Ageusia	-	21 (63.63)	20 (60.60)	20 (60.60)	0.9582
Hospitalization					
Yes	-	0 (0.0)	28 (84.85)	33 (100)	<0.0001
No	-	33 (100)	5 (15.15)	0 (0.0)	
ICU admission					
Yes	-	0 (0.0)	3 (9.09)	9 (27.27)	0.0011
No	-	33 (100)	30 (90.90)	24 (72.72)	
Ventilation support					
Yes	-	0 (0.0)	1 (3.03)	33 (100)	<0.0001
No	-	33 (100)	32 (96.96)	0 (0.0)	

SD: Standard deviation; ICU: intensive care unit.

**Table 2 viruses-15-00898-t002:** IgG antibody response to the S (S1 and S2) and N proteins in individuals with different clinical profiles of COVID-19.

Clinical Profile	Low *n* (%)	Medium *n* (%)	High *n* (%)	No Response *n* (%)	*p* Value
Subunit S1					
Asymptomatic	7 (18.9%)	13 (35.1%)	4 (10.8%)	13 (35.1%)	<0.0001
Mild	11 (33.3%)	9 (27.2%)	6 (18.1%)	7 (21.2%)	
Moderate	7 (21.2%)	6 (18.1%)	14 (42.4%)	6 (18.1%)	
Severe	2 (6.0%)	6 (18.1%)	24 (72.7%)	1 (3.0%)	
Subunit S2					
Asymptomatic	1 (2.7%)	0 (0.0%)	2 (5.4%)	34 (91.8%)	0.3222
Mild	2 (6.0%)	0 (0.0%)	0 (0.0%)	31 (93.9%)	
Moderate	1 (3.0%)	2 (6.0%)	0 (0.0%)	30 (90.9%)	
Severe	0 (0.0%)	1 (3.0%)	2 (6.0%)	30 (90.9%)	
N Protein					
Asymptomatic	12 (32.4%)	8 (21.6%)	3 (8.1%)	14 (37.8%)	<0.0001
Mild	3 (9.1%)	17 (51.5%)	11 (33.3%)	2 (6.1%)	
Moderate	5 (15.2%)	3 (9.1%)	21 (63.6%)	4 (12.1%)	
Severe	1 (3.0%)	8 (24.2%)	23 (69.7%)	1 (3.0%)	

## Data Availability

The raw data supporting the conclusions of this article will be made available by the authors without undue reservation.

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
