# Peer review of "Antibody Response to the SARS-CoV-2 Spike and Nucleocapsid Proteins in Patients with Different COVID-19 Clinical Profiles"

_viruses, 2023, doi:10.3390/v15040898_

Round 1
Reviewer 1 Report
Title:
Antibody Response to the SARS-CoV-2 Spike and Nucleocapsid Proteins in Patients with Different COVID-19 Clinical Profiles
Authors
Sinei Ramos Soares, Maria Karoliny da Silva Torres, Sandra Souza Lima, Kevin Matheus Lima de Sarges, Erika Ferreira dos Santos, Mioni Thieli Figueiredo Magalhães De Brito, Andréa Luciana Soares Da Silva, Mauro de Meira Leite, Flávia Póvoa da Costa, Marco Henrique Damasceno Cantanhede, Rosilene Da Silva, Adriana de Oliveira Lameira Veríssimo, Izaura Maria Vieira Cayres-Vallinoto, Rosimar Neris Martins Feitosa, Juarez Antônio Simões Quaresma, Tânia do Socorro Souza Chaves, Giselle Maria Rachid Viana, Luiz Fábio Magno Falcão, Eduardo Melo Santos, Antonio Carlos Rosário Vallinoto *, Andréa Nazaré Monteiro Rangel Da Silva
Submitted to section: General Virology,
SARS-CoV-2 Research in Brazil
In this study the authors analyzed IgG antibody responses to the S1, S2, and N proteins of SARS-CoV-2 in 136 individuals from June 2020 to June 2021, who had been clinically or laboratory-diagnosed with COVID-19 and were categorized as having mild, moderate, or severe disease using ELISA based assay. The authors found that the majority of patients both symptomatic and asymptomatic had IgG antibodies to the S1 and N subunit unlike the S2 subunit. They also claimed that "individuals with long COVID-19 showed a greater neutralization profile than those with symptomatology of a short duration." Hence, the authors concluded that "the levels of IgG antibodies may be related to the clinical evolution of COVID-19, with high levels of neutralizing antibodies against S1 and N in severe cases and in individuals with long COVID-19."
Major concerns
- The general concept of this study is interesting but not novel. Although the authors were aware of the limitation in the methodology used, the IgG antibody detected by the ELISA method used was consistently referred to as "neutralizing IgG" throughout the manuscript. This must be changed throughout the text, tables and figures, or the authors must conduct neutralization assays to demonstrate the neutralizing capacity of the measured or detected IgG. The authors could also show the data proving a good correlation of the methodology used with PRNT ?
- Lines 97-98: "to our understanding of the clinical status of individuals and to identifying a prognostic marker." It is not clear how this data will help in understanding the clinical status and identifying prognostic markers since asymptomatic patients had the same antibody profile.
Minor concerns:
- Line 171 & line 173 the authors referred to patients as "analyzed individuals". It is preferable to use "participants".
- In line 174 – 176 "The results showed that 87.5% of the analyzed individuals (119/136) responded to the S1 subunit and that 88.25% (120/136) responded to N. For the S1 subunit, 96.97% of the individuals with severe clinical disease (32/33), 87.88% of those with moderate disease (29/33), 93.94% of those with mild disease (31/33), and 72.97% of the asymptomatic individuals (27/37) displayed a response" the group of asymptomatic patients in table 1 was 33 while in paragraph 3.2 was 37. Could you explain this discrepancy please?
- The numbers in table 2 could be summarized as such
|
Low |
|
N (%) |
|
7 (18.92%)
|
- It is preferable to replace "Ventilatory Support" with "Ventilation Support."

Author Response
Reviewer #1
Title:
Antibody Response to the SARS-CoV-2 Spike and Nucleocapsid Proteins in Patients with Different COVID-19 Clinical Profiles
Authors
Sinei Ramos Soares, Maria Karoliny da Silva Torres, Sandra Souza Lima, Kevin Matheus Lima de Sarges, Erika Ferreira dos Santos, Mioni Thieli Figueiredo Magalhães De Brito, Andréa Luciana Soares Da Silva, Mauro de Meira Leite, Flávia Póvoa da Costa, Marco Henrique Damasceno Cantanhede, Rosilene Da Silva, Adriana de Oliveira Lameira Veríssimo, Izaura Maria Vieira Cayres-Vallinoto, Rosimar Neris Martins Feitosa, Juarez Antônio Simões Quaresma, Tânia do Socorro Souza Chaves, Giselle Maria Rachid Viana, Luiz Fábio Magno Falcão, Eduardo Melo Santos, Antonio Carlos Rosário Vallinoto *, Andréa Nazaré Monteiro Rangel Da Silva
Submitted to section: General Virology,
SARS-CoV-2 Research in Brazil
In this study the authors analyzed IgG antibody responses to the S1, S2, and N proteins of SARS-CoV-2 in 136 individuals from June 2020 to June 2021, who had been clinically or laboratory-diagnosed with COVID-19 and were categorized as having mild, moderate, or severe disease using ELISA based assay. The authors found that the majority of patients both symptomatic and asymptomatic had IgG antibodies to the S1 and N subunit unlike the S2 subunit. They also claimed that "individuals with long COVID-19 showed a greater neutralization profile than those with symptomatology of a short duration." Hence, the authors concluded that "the levels of IgG antibodies may be related to the clinical evolution of COVID-19, with high levels of neutralizing antibodies against S1 and N in severe cases and in individuals with long COVID-19."
Major concerns
- The general concept of this study is interesting but not novel. Although the authors were aware of the limitation in the methodology used, the IgG antibody detected by the ELISA method used was consistently referred to as "neutralizing IgG" throughout the manuscript. This must be changed throughout the text, tables and figures, or the authors must conduct neutralization assays to demonstrate the neutralizing capacity of the measured or detected IgG. The authors could also show the data proving a good correlation of the methodology used with PRNT ?
Unfortunately, it was not possible to use other techniques, such as PRNT, to assess the level of neutralizing antibodies, due to financial limitations. We therefore agree with your comment and in the text, tables and figures we correct and renamed what had been called “neutralization profile” for “IgG antibody response”.
- Lines 97-98: "to our understanding of the clinical status of individuals and to identifying a prognostic marker." It is not clear how this data will help in understanding the clinical status and identifying prognostic markers since asymptomatic patients had the same antibody profile.
We understand that drawing a profile with prognostic markers for a multifactorial disease is really a challenge. However, we found that in asymptomatic individuals, although all are positive for N and S, since they came into contact with the virus, the concentration of IgG antibodies for these individuals is lower than that found for critically ill patients. In this way, improving the diagnosis by measuring antibody levels and, when possible, the neutralization profile for those individuals with COVID-19, may in the future lead to better management of the same.
Minor concerns:
- Line 171 & line 173 the authors referred to patients as "analyzed individuals". It is preferable to use "participants".
We agree with the suggestion and the change was made in the text.
- In line 174 – 176 "The results showed that 87.5% of the analyzed individuals (119/136) responded to the S1 subunit and that 88.25% (120/136) responded to N. For the S1 subunit, 96.97% of the individuals with severe clinical disease (32/33), 87.88% of those with moderate disease (29/33), 93.94% of those with mild disease (31/33), and 72.97% of the asymptomatic individuals (27/37) displayed a response" the group of asymptomatic patients in table 1 was 33 while in paragraph 3.2 was 37. Could you explain this discrepancy please?
Thanks for your comment. We made a mistake and already standardized the number of individuals in the table as in the text..
- The numbers in table 2 could be summarized as such
Low
N (%)
7 (18.92%)
We agree with your suggestion and so we have adjusted the percentage values in table 2, thus reducing the space occupied by the table.
- It is preferable to replace "Ventilatory Support" with "Ventilation Support."
We agree with the suggestions and the change was made in the text.
Finally, we would like to thank in advance the reviewer for this judicious reading and suggestions.

Reviewer 2 Report
This paper describes the antibody response to spike (S1 and S2) and nucleocapsid proteins of SARS-CoV-2 in patients with different severity profiles of COVID-19. Antibody levels are measured and the potential neutralizing efficacy using a commercial ELISA kit. The authors also include data from patients with long COVID.
The finding that patients with severe disease have higher levels of antibodies has been shown in several studies. It is also known that severe disease is a risk factor for long COVID. Thus, the finding that antibody levels is higher in the long COVID patients fits nicely with this.
I do have some questions and there are things that should be clarified in this manuscript.
Reference 1, page 2, I think you can remove “National Health Commission …” in the text.
In the second paragraph on page 2, some of the detailed information about the virus seems irrelevant.
I don’t understand the meaning of the sentence starting with “Nevertheless” in line 68, page 2.
Line 74, page 2: I don’t understand that N will induce a T cell response to vaccines as most vaccines are based on spike only, but perhaps other vaccines are used in Brazil? Or are the authors talking about SARS-CoV and not SARS-CoV-2? It is a bit confusing.
Line 89, page 2: I am not sure that herd immunity is obtainable for SARS-CoV-2 https://pubmed.ncbi.nlm.nih.gov/35356987/
Lines 96-98, page 2: The authors state that the finding will contribute to decision-making for better vaccines and identify a prognostic marker. I cannot see that this is discussed later. Did their finding support these aims? The abstract, lines 43-44, page 1, includes a sentence about this, but not the full text.
Study design: did all the patients have laboratory-confirmed COVID-19? In the abstract, line 29, page 1, it says that the diagnosis was based on “clinical and/or laboratory results”, but in line 109, page 3, it says that they had confirmed infection and clinical signs. What is correct? There it also says that the 136 individuals had “clinical signs of infection”, however, 37 of the 136 are the asymptomatic group defined later. If the infections were laboratory-confirmed, did you use a PCR test or also rapid antigen tests?
Were the 59 individuals with long COVID and 33 individuals with acute COVID-19 among the 136 patients? By acute COVID, I assume that there were in the convalescent phase at the time of blood sampling? How were the 33 individuals chosen from the 77 remaining “non-long-COVID” patients? Did you try to severity match the acute and long COVID groups? I think the severity of these two different groups should have been presented (perhaps in results). It is not until the discussion line (295-298, page 9) that we find out that the long COVID patients had severe and moderate disease and the acute had mild disease and therefore not surprising that their antibody levels were different, given the authors’ previous findings.
Line 123, page 2: the blood sampling was 120 days for some of the 136 patients, but not for the long COVID patients? (90 days). Why was there a difference? Timing may be of importance due to antibody waning, perhaps the time since symptom onset can be included for all groups.
Page 4, paragraph 2.3. It says that the Kruskal-Wallis test is performed on OD-values, but what kind of test is used in Table 1 to compare the severity groups? Is that also KW?
Table 1, page 4: The number of asymptomatic patients is wrong in the first row (both according to the text in methods and the percentages of the table). You may not need so many digits in most of the percentages, that also goes for the rest of the text. I don’t know the style of this journal, but perhaps the significant p values should be labeled in bold or italics?
Figure 1: the y-axis labels are quite small. Also, there are some very small (almost not readable) p values in A-C. What do these p values indicate?
Paragraph 3.2, page 5: Figure 1C is mentioned before 1B in the text. Line 180-189: is 14.44% the percentage of all patients testing positive for S2? Figure B does not show that, as an “all” groups is not included.
Line 189, page 5: one group is now called “critically ill”. Are these the severe patients? I think the naming of groups should be consequent throughout the paper.
Table 2, page 6: I think the columns showing the number of individuals and the correspond percentages could be merged to reduce the total number of columns and making the table easier to read. Also, the number of patients in each severity group could be included somewhere. The headings of the columns are “Low”, “Middle” and “High”. In the text of page 5, it is written “moderate” and “average” neutralizing potential, does this correspond to “middle” in Table 2? On page 3 (methods), line 149-150, the neutralizing potential is called “Low”, “Medium” and “High” and “Very High”, again not the same as in names in Table 2. Was “Very High” and “High” levels combined to “High” in Table 2 or did no individual classify as “Very High”? As in Figure 1, S2 is the second antigen to be presented in the Figure/Table, but the results for nucleocapsid is presented first in the text.
Figure 2, page 7 and Fig 3, page 8. The x-axis labels are difficult to read, they are very faint. This also applies to the numbering of the y-axis. The number of individuals in each group could be added in the figure legends.
Line 208-209: For S1 there was a significant difference between the severe and moderate groups. This is not indicated in Fig. 2A.
Line 224, page 7: Figure 2B shows the S2 results, not N. Consequently, Figure 2C is not referenced in the text.
Discussion, page 8. Is paragraph 2, lines 238-245, what is the relevance of this information for the present study?
Lines 272-276, page 9, results?
Lines 290-303: As mentioned previously, I think the severity profile of the long COVID and acute COVID groups should have been mentioned earlier. Moreover, according to the CDC, severe COVID-19 is a risk factor for long COVID https://www.cdc.gov/coronavirus/2019-ncov/long-term-effects/index.html even though it can also occur in people with mild disease.
Line 307, page 10: What is known about the variants that the included patients were ill from? Since the samples were collected between June 2020 to June 2021, some may have been ill from Wuhan and some from Alpha? Did you study if there were differences within the severity groups based on when they were ill (when in the year and what virus type that was circulating at the time of illness)?
Author Response
Reviewer #2
This paper describes the antibody response to spike (S1 and S2) and nucleocapsid proteins of SARS-CoV-2 in patients with different severity profiles of COVID-19. Antibody levels are measured and the potential neutralizing efficacy using a commercial ELISA kit. The authors also include data from patients with long COVID.
The finding that patients with severe disease have higher levels of antibodies has been shown in several studies. It is also known that severe disease is a risk factor for long COVID. Thus, the finding that antibody levels is higher in the long COVID patients fits nicely with this.
I do have some questions and there are things that should be clarified in this manuscript.
Reference 1, page 2, I think you can remove “National Health Commission …” in the text.
We agree with the suggestion and the information has already been removed from the text.
In the second paragraph on page 2, some of the detailed information about the virus seems irrelevant.
We are grateful for the suggestion, but we understand that it is better to keep the information about the viral structure so that since it supports the design of the serological test used for this research, where the IgG response profile for the N and S proteins is evaluated.
I don’t understand the meaning of the sentence starting with “Nevertheless” in line 68, page 2.
We agree with the suggestion and the word has already been removed from the text.
Line 74, page 2: I don’t understand that N will induce a T cell response to vaccines as most vaccines are based on spike only, but perhaps other vaccines are used in Brazil? Or are the authors talking about SARS-CoV and not SARS-CoV-2? It is a bit confusing.
Thank you for your comment and to a better understanding, in the text we do not aim to talk about vaccine response. According to your question, we have modified the information in the text.
Line 89, page 2: I am not sure that herd immunity is obtainable for SARS-CoV-2 https://pubmed.ncbi.nlm.nih.gov/35356987/
Thanks for your observation. Yes, we agree that herd immunity seems to be difficult to achieve for COVID-19, once getting the disease offers some natural protection for reinfection however, even with antibodies, you could get COVID-19 again including evolving for severe complications. Into this perspective and considering your comment, the text was modified for a better clarity.
Lines 96-98, page 2: The authors state that the finding will contribute to decision-making for better vaccines and identify a prognostic marker. I cannot see that this is discussed later. Did their finding support these aims? The abstract, lines 43-44, page 1, includes a sentence about this, but not the full text.
We agree with your statement that our work is not enough to improve vaccines or to search for prognostic markers. However, any study that brings information about the response profile to COVID-19 in different population is valuable for understanding the disease. Considering that we do not intend here to discuss the vaccine or infer prognostic markers, the information was removed from the abstract.
Study design: did all the patients have laboratory-confirmed COVID-19? In the abstract, line 29, page 1, it says that the diagnosis was based on “clinical and/or laboratory results”, but in line 109, page 3, it says that they had confirmed infection and clinical signs. What is correct? There it also says that the 136 individuals had “clinical signs of infection”, however, 37 of the 136 are the asymptomatic group defined later. If the infections were laboratory-confirmed, did you use a PCR test or also rapid antigen tests?
Thanks for your comment. All individuals participating in the research had a confirmed laboratory diagnosis by ELISA, so all this information was corrected in the abstract.
Were the 59 individuals with long COVID and 33 individuals with acute COVID-19 among the 136 patients? By acute COVID, I assume that there were in the convalescent phase at the time of blood sampling? How were the 33 individuals chosen from the 77 remaining “non-long-COVID” patients? Did you try to severity match the acute and long COVID groups? I think the severity of these two different groups should have been presented (perhaps in results). It is not until the discussion line (295-298, page 9) that we find out that the long COVID patients had severe and moderate disease and the acute had mild disease and therefore not surprising that their antibody levels were different, given the authors’ previous findings.
Yes, we used the same patients with mild, moderate and severe disease to see if they have signs of long COVID. We had criteria for classifying these individuals with ''long COVID'' and ''acute COVID'' and these criteria was based on the duration of symptoms. The persistence of symptoms for more than 90 days was classified as long COVID and those individuals into the severe and moderate group were those that fell under the long COVID profile.
In regarding to severity match between acute and long Covid, we added the following sentence in the discussion.
“This result probably is more related to severity during the acute Covid-19 period than to differences in response between long and acute Covid-19. Future analyzes addressing the pairing of clinical status between the groups may elucidate this issue.”
Line 123, page 2: the blood sampling was 120 days for some of the 136 patients, but not for the long COVID patients? (90 days). Why was there a difference? Timing may be of importance due to antibody waning, perhaps the time since symptom onset can be included for all groups.
We understand that time is an important factor for antibody waning. Into this perspective we did statistical analyses in order to see if the time of blood sampling could impact our results but we did not find statistical significance between the groups.
Page 4, paragraph 2.3. It says that the Kruskal-Wallis test is performed on OD-values, but what kind of test is used in Table 1 to compare the severity groups? Is that also KW?
As described on page 4, in descriptive statistics, the statistical test used to compare characteristics such as: age, gender, symptoms and hospitalization characteristics (Table 1) in relation to severity was the G and KW test. We reaffirm this in the body of the text for a better interpretation of the results.
Table 1, page 4: The number of asymptomatic patients is wrong in the first row (both according to the text in methods and the percentages of the table). You may not need so many digits in most of the percentages, that also goes for the rest of the text. I don’t know the style of this journal, but perhaps the significant p values should be labeled in bold or italics?
We appreciate your placements and with this, we alter the values that would be wrong and standardize the significant values in bold.
Figure 1: the y-axis labels are quite small. Also, there are some very small (almost not readable) p values in A-C. What do these p values indicate?
These p values indicate that there is a significant difference in the total IgG response to the SARS-CoV-2 S and N proteins in individuals with different clinical profiles of COVID-19. We increase the size of the font to facilitate reading.
Paragraph 3.2, page 5: Figure 1C is mentioned before 1B in the text. Line 180-189: is 14.44% the percentage of all patients testing positive for S2? Figure B does not show that, as an “all” groups is not included.
We thank you for your cooperation and we have changed the citation order of the figure, lines 184-188. The value of 14.44% is the percentage of reactive tests for S2. Figure 1B, shows that among the 136 individuals analyzed, only 14.44% reacted to S2, note that highlighted in yellow, only a small percentage of patients, regardless of severity, reacted to the S2 subunit. (PROFA melhore essa resposta que aqui já fiquei irritada com esse revisor kk)
Line 189, page 5: one group is now called “critically ill”. Are these the severe patients? I think the naming of groups should be consequent throughout the paper.
We agree and thanks for your observation and the correction already was made in the text.
Table 2, page 6: I think the columns showing the number of individuals and the correspond percentages could be merged to reduce the total number of columns and making the table easier to read.
We thank you for your suggestion and table 2 was modified.
The headings of the columns are “Low”, “Middle” and “High”. In the text of page 5, it is written “moderate” and “average” neutralizing potential, does this correspond to “middle” in Table 2?
We agree and this was corrected in the text.
On page 3 (methods), line 149-150, the neutralizing potential is called “Low”, “Medium” and “High” and “Very High”, again not the same as in names in Table 2. Was “Very High” and “High” levels combined to “High” in Table 2 or did no individual classify as “Very High”?
Given the detection limit of our ELISA equipment, ODs greater than 3.0 couldn’t be read and therefore some of our patients cannot be classified within the “very high” profile”.
As in Figure 1, S2 is the second antigen to be presented in the Figure/Table, but the results for nucleocapsid is presented first in the text.
We agree and this was corrected in the text.
Figure 2, page 7 and Fig 3, page 8. The x-axis labels are difficult to read, they are very faint. This also applies to the numbering of the y-axis. The number of individuals in each group could be added in the figure legends.
Line 208-209: For S1 there was a significant difference between the severe and moderate groups. This is not indicated in Fig. 2A.
As per your recommendations, we changed the axes and legends of figure 2.
Line 224, page 7: Figure 2B shows the S2 results, not N. Consequently, Figure 2C is not referenced in the text.
We agree and this was corrected in the text.
Discussion, page 8. Is paragraph 2, lines 238-245, what is the relevance of this information for the present study?
We agree and the information was removed from the text.
Lines 272-276, page 9, results?
The sentence was removed.
Lines 290-303: As mentioned previously, I think the severity profile of the long COVID and acute COVID groups should have been mentioned earlier. Moreover, according to the CDC, severe COVID-19 is a risk factor for long COVID https://www.cdc.gov/coronavirus/2019-ncov/long-term-effects/index.html even though it can also occur in people with mild disease.
We described on methodology the clinical parameters considered for long COVOD-19: For acute COVID-19, individuals were classified as mild, moderate or severe COVID-19 following the criteria of the WHO (19) with regard to the occurrence of clinical symptoms of fever, cough, fatigue, anorexia, shortness of breath, myalgia, pneumonia, acute respiratory distress syndrome and oxygenation impairment.The main symptoms reported in the long COVID-19 group were headache, depression, anxiety, insomnia, tingling in extremities, mild cognitive disorder, tiredness, dyspnea, weakness, pain, anosmia, and ageusia.
Line 307, page 10: What is known about the variants that the included patients were ill from? Since the samples were collected between June 2020 to June 2021, some may have been ill from Wuhan and some from Alpha? Did you study if there were differences within the severity groups based on when they were ill (when in the year and what virus type that was circulating at the time of illness)?
Unfortunately, we do not have any information about the variant that infected our patients.
Finally, we would like to thank in advance the reviewer for this judicious reading and suggestions.

Round 2
Reviewer 1 Report
Title:
Antibody Response to the SARS-CoV-2 Spike and Nucleocapsid Proteins in Patients with Different COVID-19 Clinical Profiles
Authors
Sinei Ramos Soares, Maria Karoliny da Silva Torres, Sandra Souza Lima, Kevin Matheus Lima de Sarges, Erika Ferreira dos Santos, Mioni Thieli Figueiredo Magalhães De Brito, Andréa Luciana Soares Da Silva, Mauro de Meira Leite, Flávia Póvoa da Costa, Marco Henrique Damasceno Cantanhede, Rosilene Da Silva, Adriana de Oliveira Lameira Veríssimo, Izaura Maria Vieira Cayres-Vallinoto, Rosimar Neris Martins Feitosa, Juarez Antônio Simões Quaresma, Tânia do Socorro Souza Chaves, Giselle Maria Rachid Viana, Luiz Fábio Magno Falcão, Eduardo Melo Santos, Antonio Carlos Rosário Vallinoto *, Andréa Nazaré Monteiro Rangel Da Silva
Submitted to section: General Virology,
SARS-CoV-2 Research in Brazil
In this study the authors analyzed IgG antibody responses to the S1, S2, and N proteins of SARS-CoV-2 in 136 individuals from June 2020 to June 2021, who had been clinically or laboratory-diagnosed with COVID-19 and were categorized as having mild, moderate, or severe disease using ELISA based assay. The authors found that the majority of patients both symptomatic and asymptomatic had IgG antibodies to the S1 and N subunit unlike the S2 subunit. They also claimed that "individuals with long COVID-19 showed a greater neutralization profile than those with symptomatology of a short duration." Hence, the authors concluded that "the levels of IgG antibodies may be related to the clinical evolution of COVID-19, with high levels of neutralizing antibodies against S1 and N in severe cases and in individuals with long COVID-19."
Thank you for taking the time to revise the manuscript and for the extensive literature review to support your results.
Major concerns
- I believe the fundamental premise of this work is interesting but not original, and the use of an assay with low specificity compromises the validity of the conclusions. There is no need to use the PRNT assay but a higher specificity test.
Minor concerns: There are still several grammatical mistakes in the text.
Author Response
Dear Reviewer,
Thank you for your suggestion but, unfortunately, we no longer have the financial resources to purchase other tests, as funding for the present study ended in July 2022. The test DiaPro's ELISA used in the present study is not of low specificity. According to the manufacturer’s reports, the assay specificity, evaluated examining more than one hundred plasma collected before and after the outbreak of COVID-19, first tested negative for IgG on the specific DiaPro's ELISA, reached an overall value of almost 100%. No false positive reactions were observed. Additionally, the same performance was showed by Marlet et al. (Journal of Clinical Virology 132 (2020) 104633) in an analysis of Clinical performance of four immunoassays for antibodies to SARS-CoV-2.
Furthermore, at the time we carried out the study it was what was available for differential analysis of anti-S1, anti-S1 and anti-N antibody response. We believe that although the results may not be 100% original, they certainly add information about the humoral immune response in acute and long-term Covid-19. In addition, the present work demonstrated that besides the Spike protein, the N protein is also immunogenic, reinforcing the importance of rethinking vaccine approaches not only for Spike expression. Also, these results will be important to stimulate new studies for more robust methodologies and diagnostic tests . In this sense, the authors would like to obtain your understanding and agreement in the publication of the presented data.
Regarding the need for grammatical revision, the article was again resubmitted to grammar review by the American Journal of Experts.
Once again, the authors would like to thank you for your important suggestions that certainly improved the quality of the manuscript.